# Higher Virulence of *Diplodia seriata* Isolates on Vines of cv. Cabernet Sauvignon Associated with 10-Year-Old Wood Compared to Young Tissue

**DOI:** 10.3390/plants12162984

**Published:** 2023-08-18

**Authors:** Alejandra Larach, Paulina Vega-Celedón, Eduardo Salgado, Aldo Salinas, Natalia Riquelme, Diyanira Castillo-Novales, Paulina Sanhueza, Michael Seeger, Ximena Besoain

**Affiliations:** 1Escuela de Agronomía, Facultad de Ciencias Agronómicas y de los Alimentos, Pontificia Universidad Católica de Valparaíso, Casilla 4-D, Quillota 2260000, Chile; alejandra.larach@pucv.cl (A.L.);; 2Laboratorio de Microbiología Molecular y Biotecnología Ambiental, Departamento de Química & Centro de Biotecnología Dr. Daniel Alkalay Lowitt, Universidad Técnica Federico Santa María, Avenida España 1680, Valparaíso 2390123, Chile

**Keywords:** plant–pathogen interactions, grape trunk diseases, GTD, Botryosphaeria dieback, pathogenicity, tissue age, virulence, *Vitis vinifera*, cabernet sauvignon

## Abstract

Botryosphaeria dieback (BD) occurs in young and old plants. In the field, the prevalence and severity of the disease increase proportionally with the age of vineyards. Among the pathogens that cause BD, *Diplodia seriata* is the most prevalent species in Chile and other countries with a Mediterranean climate. To date, no information is available on the susceptibility of adult wood to infection by this pathogen since most of the pathogenicity tests have been carried out on 1- or 2-year-old shoots or detached canes. Therefore, a pathogenicity test was carried out on plants under field conditions, with inoculations in 1-year-old shoots and 2- and 10-year-old wood in grapevine cv. Cabernet Sauvignon. A pathogenicity test was carried out with two isolates of *D. seriata*. The results for the plants show that *D. seriata* was significantly more aggressive on the 10-year-old than on the one- or two-year-old tissue, where the lesions were 4.3 and 2.3 cm on average, respectively. These results were compared with the lesions obtained from two-year-old canes after the isolates were activated in grape berries. Also, the Chilean isolates of *D. seriata* were compared phylogenetically with those from other countries, and no major differences were found between them. Our results are consistent with the damage observed in the field, contributing to the knowledge of the epidemiology of this disease in Mediterranean climates. In the future, the effect observed in cv. Cabernet Sauvignon with *D. seriata* on virulence at different tissue ages should be tested for other BD-causing agents and wine varieties.

## 1. Introduction

Grapevine trunk diseases (GTDs) are the leading cause of vineyard deterioration, causing a decrease in the viability of vineyards, an increase in production costs, and major economic losses in the wine industry [1,2,3,4,5,6]. The three main GTDs are esca disease, Eutypa dieback, and Botryosphaeria dieback (BD), which generally attack the structural parts of plants [2,6,7,8,9,10]. BD is present on all continents where wine or table grapes are grown [2,5,7,11,12,13,14,15,16]. Regarding the pathogens associated with BD, 26 species have been identified [10], mainly within the genera *Botryosphaeria*, *Diplodia*, *Lasiodiplodia*, and *Neofusicoccum* [7,10,11,17]. Some genera are more common in different climates, for example, *Diplodia* in temperate regions or those with cold winters and *Lasiodiplodia* in tropical and subtropical areas [18,19,20,21,22,23,24]. *Diplodia seriata* is the most prevalent species in vineyards in Central Chile [5,15,22] as well as in other parts of the world, such as New South Wales [25,26,27], Western Australia [28], California [29], and South Australia [26].

BD causes essential losses in the production of grapes for wine, with a decrease from 30 to more than 50% of production depending on the severity of the disease [5]. The loss of production is associated with the damage it causes to the vines due to the death of the wood where fruit develops, such as cankers in the wood, dead spurs, dead arms, and brown discolorations in the vascular area [2,5,7,15,16,30,31,32,33,34]. In the field, the prevalence and severity of (BD) increase proportionally with the age of the vineyards [5,6,7,10,15,33]. 

Considering physiological and molecular studies carried out with Botryosphaeriaceae, Bernard-Gellon et al.’s [35] work with the callous tissue of *V. vinifera* cv. Chardonnay found that *N. parvum* produced more extracellular proteins in higher concentrations than *D. seriata*, and equivalent concentrations of proteins secreted by *N. parvum* were more aggressive than those of *D. seriata* in producing necrosis and induced more grapevine defense genes. Morales-Cruz et al. [36] found that *D. seriata* has fewer genes associated with CAZymes enzymes, enzymes associated with secondary metabolism, active enzymes, and cytochrome P450 than *N. parvum.*

However, many available pathogenicity studies have been conducted on callus tissue on one-year-old shoots and two-year-old canes. Another critical issue to consider is the susceptibility and importance of some varieties for the wine industry. In this case, Cabernet Sauvignon is the most planted and susceptible variety in Chile, and it is appreciated for good quality wines [37].

For this reason, in this study, a pathogenicity test was carried out with five Chilean isolates of *D. seriata* in 2-year-old detached canes of *Vitis vinifera* cv. Cabernet Sauvignon, and with the two most representative isolates, a pathogenicity test was carried out on vine plants under field conditions, with inoculations in 1-year-old shoots, 2-year-old wood, and 10-year-old wood of grapevine. 

## 2. Results

### 2.1. Pathogenicity Tests on Berry Grapes

All isolates caused circular soft rot around the point of inoculation in grape berries, with lesion diameters between 2.1 and 2.9 cm and the presence of grayish-white mycelium around the point of inoculation (Figure 1a,b). The largest lesions were caused by isolate 1549 (2.9 cm), followed by isolate 2142 (2.6 cm), and then by isolates 1472 (2.1), 2120 (2.2 cm), and 2183 (2.1 cm). The pathogen was recovered from the area of advancement of the lesions, and *D. seriata* was morphologically identified. The controls (uninoculated berry grape) did not show damage (Figure 1c), thus corroborating that all *D. seriata* isolates were pathogenic in berry cv. Red Globe.

### 2.2. Pathogenicity Tests in Detached Canes (Activated Isolates)

All isolates produced canker lesions on inoculated canes, with lengths between 1.4 and 3.0 cm (Figure 2a), and isolates PUCV 2120 and PUCV 2183 causing the longest cankers. Isolates PUCV 1472, PUCV 2120, and PUCV 2183 also produced vascular discoloration, which was expressed up and down from the point of inoculation. The vascular lesions presented variable lengths between 2.6 and 13.3 cm (Figure 2b), and as with the cankers, isolates PUCV 2120 and PUCV 2183 were the ones that caused the most significant lesions. It is also important to note that the average size of the vascular lesions was 4.1 times longer than that of the canker lesions. The vascular lesions and cankers were brown and dark brown, respectively; the negative controls (noninoculated canes) did not present damage (no canker and no vascular discoloration) (Figure 2c). From the lesions produced by the isolates, the pathogen was recovered, and *D. seriata* was morphologically identified. With the results obtained in detached cane assays, isolates PUCV 2120 and PUCV 2183 were selected for the field trial.

### 2.3. Pathogenicity Trial in the Field

Both isolates produced significant vascular lesions compared to the controls (T0 and T00), as shown in Figure 3. In young tissue (one-year-old shoots and two-year-old wood), the vascular lesions were significantly smaller than those in older tissue (10-year-old wood) (Figure 3 and Figure 4a). The vascular lesions developed in the 10-year-old wood were 4.2 and 4.3 cm for the PUCV 2183 and PUCV 2120 isolates, respectively, with no significant differences in the aggressiveness among the *D. seriata* isolates evaluated. In young tissue, the vascular lesions caused by *D. seriata* isolates ranged from 1.9 to 2.7 cm, with no differences between the tissue ages (one and two years old) or between the isolates (PUCV 2120 and PUCV 2183). Only from inoculated plants was the pathogen recovered from young and old tissue. This result shows that the vines used in this trial were healthy and that the lesions presented are attributable to the inoculations with *D. seriata* isolates. No plants showed foliar symptoms (either inoculated or noninoculated) (Figure 4b).

### 2.4. Maximum Parsimony Analysis

The concatenated ITS and BT phylogenetic analysis included 66 *D. seriata* isolates, all obtained from *V. vinifera (*Figure 5). The replicate trees (in percentages) of the associated taxa clustered in the bootstrap test (1000 replicates) can be seen next to the branches. The tree was obtained using the tree–bisection–regrafting (TBR) algorithm. The analyses contained 802 nucleotides and involved 79 nucleotide sequences. 

According to the results of the phylogenetic analysis, there were no major differences in the *D. seriata* isolates between the ITS and BT segments (Appendix A). However, there were differences in the virulence among the isolates, even after being activated in fruit, which is not attributable to the place of origin or the time of collection (autumn or spring) but is associated with the year of collection.

## 3. Discussion

This study reports, for the first time, a comparison of the pathogenicity of the fungus *D. seriata* on the tissues of different ages in cv. Cabernet Sauvignon and show greater aggressiveness in the old tissue under field conditions. Of the isolates obtained from diseased tissue, *D. seriata* is the most widely distributed in New South Wales, Australia, California (USA), Catalonia (Spain), and Central Chile [5,6,15,17,23,25,26,28,38,39]. Based on the works mentioned above, most isolates of *D. seriata* were obtained from symptomatic old wood tissue. However, pathogenicity tests for *D. seriata* are mainly carried out on young tissue (shoots, one- or two-year-old canes, or young tissue on old plants). In these cases, *D. seriata* virulence was weak or less significant than that of other *Botryosphaeriaceae* species, such as *Lasiodiplodia theobromae*, *L. viticola*, *Neofusicoccum parvum*, or *Diplodia mutila* [8,33,40,41], or when compared to other causal agents of GTDs, such as *Eutypa lata* or *N. parvum* [42,43]. The difference in virulence could be due to several factors, including climatic conditions, varietal susceptibility of the vine, age of the host plant tissue, type of inoculation performed, geographic origin of the isolates, and period of experiment development, among others [7,31,32,43,44]. Also, it is important to study the microbiome associated with plants and its influence on disease development [45].

In our work, the pathogenicity of *D. seriata* isolates was verified in plants, with inoculations in young tissue (one- or two-year-old tissue age) and in old tissue (10-year-old wood) in vine plants. The vascular lesions developed by *D. seriata* on old tissue (10-year-old wood) were more than twice as long as those in the young tissue. Our results coincide with those described by Morales et al. [15] for old table grape plants in a commercial vineyard. When inoculating shoots < one year, canes > five years old, and mature arms in 25-year-old vines in commercial vineyards, *D. seriata* increased the damage by more than 40% in the mature arms with respect to the other two ages evaluated, contrary to *D. mutila.* Additionally, our results are consistent with those reported in naturally affected vineyards. In the field, symptoms of BD increase with the age of the vineyard, both in incidence and severity [5,12,15,30]. The extension of the advancement of this pathogen in the wood can be explained by the microenvironmental theory, where pathogen reaction zones can alone explain the progression of fungi within tree trunks, as fungi follow dehydrated, oxygen-rich zones that occur along the reaction zones near the wounds [46]. The high moisture content and the associated aeration restriction limit the activity of mycelial fungi in felled wood. Additionally, the most significant damage in old tissue can be explained concerning the plant microbiome, which evolves over time and with plant tissues. Microbiome diversity is higher in young vines [1,47,48,49] and in visually healthy tissues than in diseased tissues [50].

In our study, when comparing the results of the pathogenicity tests on two-year-old tissue, detached cuttings, or plant canes (attached), a difference in the severity of the damage of *D. seriata* was obtained, where it was more aggressive in detached canes (inoculated with mycelium plug) than in adhered canes (inoculated with conidia). Vascular lesions were up to 12 cm in detached canes, while in attached canes, they were up to 5 cm. This may be due to the type of inoculum (mycelium or conidia) and/or the state of the tissue (detached or attached). Regarding the type of inoculum, our results are contrary to those reported by Moral et al. [51], where inoculation with conidia of *Botryosphaeria dothidea* resulted in a more severe disease than inoculation with a mycelium plug in olive fruits. Concerning the state of the tissue, the literature describes that during grapevine infection, a series of molecules are activated that protect against the pathogen’s spread [50]. The first active defense is the formation of tylosis within the vessels. Responses to GTD fungi are thought to occur due to PAMP-activated immunity. Phytoalexins, compounds of the phenylpropanoid pathway, such as resveratrol or viniferin, are found in higher amounts in tissues after GTD fungal infection [52,53,54,55,56]. Phytoalexins can inhibit fungal growth and colonization, block some metabolites produced by fungi during infection, or interfere with oxidation–reduction reactions [57]. Due to the loss of xylem function and decreased hydraulic conductivity, which induce fungal colonization, active plant defense associated with parenchyma rays can prevent this from occurring [58]. In this work, it would have been important to measure the hydraulic conductivity of these tissues and the presence of phenols or phytoalexins specific to the vine, such as those mentioned above.

Reveglia et al. [59] demonstrated that the *Botryosphaeriaceae* family associated with BD in grapevine leaves generates secondary metabolites. The study by Martos et al. [60] pointed out that *D. seriata* can produce secondary metabolites with phytotoxic properties, dependent on acidic pH substrates, among other possible mechanisms related to its virulence. This could explain the lower virulence variability of *D. seriata* isolates when inoculated on fruits rather than on canes, although this requires further study. It is important to note that storage in culture can cause the loss of virulence or pathogenicity due to nutrient restriction for a prolonged period [61,62,63]. In addition to the active defense implemented by the vines, the anatomy of the wood could also be essential. The density of parenchymal rays and their arrangement in space could enhance active responses to infection [64] and the diameter of the vessels could also be a factor [58,65,66]. In contrast, Amponsah et al. [67] showed a faster rate of germination of *N. luteum* conidia on attached and wounded shoots and leaves than on detached and unwounded leaf surfaces. This may be due to the ability of *Neofusicoccum* species to attack in spring and summer time, which is associated with damage to the shoots and leaves, unlike *D. seriata*, which mainly affects woody tissue.

Furthermore, in other *Botryosphaeriaceae* species, *B. dothidea* in this case, the physiological characterization of the isolates that cause Dalmatian disease in olive showed that the optimal temperature was 26 °C for mycelium growth and 30 °C for the germination of conidia [51], which is also the optimal temperature for the conidia of *D. seriata* isolated from *V. vinifera* [68]. Both factors, the type of inoculum and the temperature, can influence the aggressiveness of the symptoms, which should be studied for *D. seriata*. The difference in virulence observed in our work was also reported by Elena et al. [44], who inoculated with mycelium different isolates of *D. seriata* in detached canes and two-year-old potted plants cv. Tempranillo. In this case, the differences were attributed to the type of host from which each isolate was obtained. The ITS and BT sites analyzed in this work showed no significant nucleotide variations between the Chilean isolates and the reference ones (Appendix A), which came from California (USA) and Baja California (Mexico). This coincides with the results found by Urbéz-Torres et al. [29,41], where low variation was detected in the ITS and BT sequences of *D. seriata* from different geographical origins concerning other pathogens causing BD, such as *L. theobromae*. Thus, *D. seriata* is considered a more cosmopolitan pathogen.

## 4. Materials and Methods

### 4.1. Chemicals, Reagents, and Culture Media

The reagents and culture media used were sodium hypochlorite (SMF Ltda., Santiago, Chile), agar papa dextrose acidulated (APDA): 20 g L^−1^ granulated agar (Algas Marinas S.A., Pontevedra, Spain), 20 g L^−1^ mashed potatoes (Nestlé S.A., Vevey, Switzerland), 22 g L^−1^ glucose (Vimaroni S.A., Valparaíso, Chile), 7 drops of citric acid (Merck S.A., Darmstadt, Germany), and 1 L of sterile distilled water (SDW). The sterile distilled water was obtained by using a distiller (Pobel, Madrid, Spain) and autoclave (Zonkia, Hefei, China). The parafilm used was obtained from Bemis, Sheboygan Falls, WI, USA.

### 4.2. Fungal Isolates

Five isolates of *D. seriata* were used (Table 1). The isolates belong to the fungal collection of the Phytopathology Laboratory, School of Agronomy, PUCV. The PUCV 1472, PUCV 1549, PUCV 2120, PUCV 2142, and PUCV 2183 isolates were previously obtained from plants with cankers and vascular lesions from commercial vineyards in the wine-growing zone of central Chile, all with a history of BD that was previously molecularly identified [5]. The isolates were classified based on a phylogenetic analysis of the concatenated ITS-BT. Multilocus phylogenetic analysis was performed using maximum parsimony (MP) in MEGA X [69]. Bootstrap values were calculated using 1000 replicates, yielding the MP tree using the tree bisection and reconnection algorithms. The tree was rooted with *Neofusicoccum parvum* strain CBS 110302, and other reference isolates of different isolates of *D. seriata* and *D. mutila* were used (Appendix A). The retention index, rescaled consistency index, and homoplasy index were calculated using MEGA X.

### 4.3. Pathogenicity Tests on Berry Grapes

The inoculations were performed on fresh berry grape cv. Red Globe of *Vitis vinifera.* The berry grapes were obtained in the summer of 2021 from the experimental vineyards of Estación La Palma of Universidad Católica de Valparaíso (PUCV) and immediately used for the test.

#### 4.3.1. Fungal Isolates and Inoculation

For the inoculation, the berry grapes were disinfected with 1% sodium hypochlorite for 30 seg and then triple-washed in sterile distilled water. A 5-mm-diameter inoculum of a 5-mm-diameter mycelial plug from a 6-day-old growth in APDA culture was inoculated in the middle of the berry through a previously made wound with a sterile needle. The inoculated area was covered with parafilm. The pathogenicity assay was performed with five isolates (Table 1). The control consisted of noninoculated berry grapes, with only an APDA plug. The berry grapes were incubated for 15 days in an individual humid chamber at 23 °C. 

#### 4.3.2. Damage Assessment and Pathogen Recovery

After the incubation of the berry grapes, fruit rot was measured around the point of inoculation. The diameter of the lesions was considered. Tissue samples were taken from the zone of advancement of the lesions, disinfected with sodium hypochlorite for 5 seg, triple-washed in ADE, and cultivated in APDA. The plates were incubated for 7 d at 24 °C.

### 4.4. Pathogenicity Tests on Detached Canes 

The inoculations were performed on healthy detached 2-year-old canes (15 cm in length) of *Vitis vinifera* cv. Cabernet Sauvignon. The canes were obtained in the winter of 2020 from an experimental vineyard of the Experimental Station La Palma of Pontifical Catholic University of Valparaíso (PUCV) and maintained at 5 °C for three months before use [65].

The pathogenicity assay was performed with five isolates (Table 1), previously activated in grape berry cv. Red Globe plants. Detached canes were disinfected (1% sodium hypochlorite for 5 min and 95% ethanol for 30 s) and then triple-washed in SDW, according to Morales et al. [15], and an inoculum of 5-mm-diameter mycelial plugs from a 6-day-old APDA culture was inoculated using an oblique cut made in the bark with a sterile scalpel in the middle of the canes. The inoculated area was covered with parafilm [15,47]. Three detached 2-year-old canes were used for each isolate, and three were used for the control (noninoculated canes, only APDA plug). The assay was performed in duplicate. The canes were incubated for 90 days in an individual humid chamber at 23 °C.

#### Damage Assessment and Pathogen Recovery

After three months of incubation, the vascular lesions and canker lengths of the canes were measured up and down from the point of inoculation. For the data analysis, the total length of the lesions was considered. Tissue samples were taken from the zone of advancement of the lesions, disinfected with 1% sodium hypochlorite for 5 s, triple-washed in SWD, and cultivated in APDA.

### 4.5. Pathogenicity Trial in the Field

The experiment was carried out on 10-year-old ungrafted cv. Cabernet Sauvignon plants in an experimental vineyard at the Experimental Station La Palma of the Pontifical Catholic University of Valparaíso (PUCV). The selected vineyard did not present previous GTD symptoms, and it was trained to the bilateral cordon system, with the spur pruned to 4 buds. The inoculations were performed on tissues of three different ages: 1-year-old shoots, 2-year-old wood, and 10-year-old wood, with each inoculation/age on a different plant.

#### 4.5.1. Fungal Isolates and Inoculation

Two isolates of *D. seriata,* PUCV 2120 and PUCV 2183 (Table 1), previously selected in pathogenicity tests on two-year-old detached canes, were used for the field experiment. The inoculation was carried out in July 2021, with 50 µL of inoculum at 1 × 10^4^ spores × mL^−1^ on fresh cuts on 1-year-old shoots, 2-year-old wood, and 10-year-old wood (arm) (Figure 6). The cuts in the 1-year-old shoots and 2-year-old wood were made with pruning shears, and in 10-year-old wood with a chainsaw (Stihl, Waiblingen, Germany). Spore suspensions were prepared according to Úrbez-Torres et al. [38] and Larach et al. [5]: 5-day-old mycelial plugs of each isolate were placed in Petri dishes containing 2% agar water and needle-autoclaved pine. The plates were incubated in a chamber at room temperature (19–21 °C) under near-ultraviolet light (λ = 320 nm) until pycnidium production and conidium development. Mature pycnidia were crushed in sterile distilled water, and the solution obtained was filtered through a sterilized cheesecloth. Five plants were used for each isolate/tissue age, with one inoculation per plant. Five absolute control plants (without cutting or inoculation, T00) and five control plants (cut plus sterile distilled water, T0) were used.

#### 4.5.2. Damage Assessment and Pathogen Recovery

Five months after the inoculation of the tissue, the vascular lesions and canker lengths were recorded. For this, the damage produced in each tissue/age was evaluated in the field, and the plant material was cut to measure the lesions. The assay was performed in duplicate. To recover the fungus, tissue samples were taken from the zone of advancement of the lesions, disinfected, and cultivated in APDA.

### 4.6. Statistical Analysis

The pathogenicity test on the berry grapes was distributed according to a completely randomized design. One berry grape was the experimental unit, and five replicates were used for each treatment. The results were analyzed for variance, and the means were compared according to Tukey’s test (*p* ≤ 0.05).

The pathogenicity test on the detached canes was distributed according to a completely randomized design. One cane was the experimental unit, and three replicates were used for each treatment. The results were analyzed for variance, and the means were compared according to Tukey’s test (*p* ≤ 0.05).

In the field, the plants of the pathogenicity trial were distributed according to a completely randomized design. One plant was the experimental unit, and five repetitions were used. An analysis of variance of two factors (tissue age and isolates) was performed, and no interaction of the factors was obtained, so a factor analysis was performed for the significant factor (tissue age) using Tukey’s test (*p* ≤ 0.05).

## 5. Conclusions

In this study, we observed that *D. seriata* is more aggressive in old tissue of grapevine plants cv. Cabernet Sauvignon than in young tissue, verifying the pathogenicity of *D. seriata* and its role as a causal agent of the disease (BD). These results contribute to the knowledge of the epidemiology of this disease in Mediterranean climates. Additionally, the pathogenicity tests carried out to select the isolates for pathogenicity testing in the field showed different results for vegetative tissue and berry grapes, highlighting the importance of the type of tissue in the pathogenicity and the virulence of the *D. seriata* pathogen. This work should be continued by employing other BD pathogens and different wine varieties.

## Figures and Tables

**Figure 1 plants-12-02984-f001:**
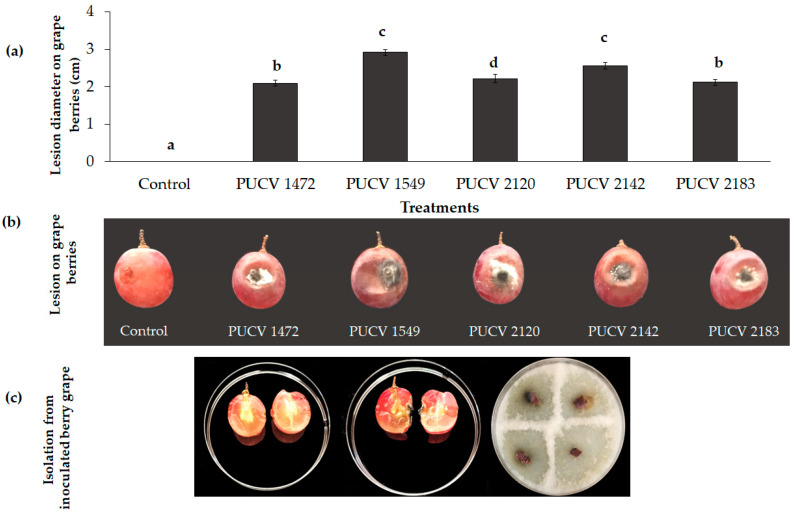
Soft rot lesions caused by *D. seriata* on *Vitis vinifera* cv. Red Globe. (**a**) Mean lesion diameter on berries inoculated with different isolates of *D. seriata*. Different letters indicate significant differences after ANOVA according to Tukey’s test (*p* ≤ 0.05). (**b**) Circular and sunken rot around the point of inoculation with the different isolates. (**c**) Left to right: control berry grapes (noninoculated), internal damage of inoculated berry grapes, and recovery of *D. seriata* from inoculated berry grapes.

**Figure 2 plants-12-02984-f002:**
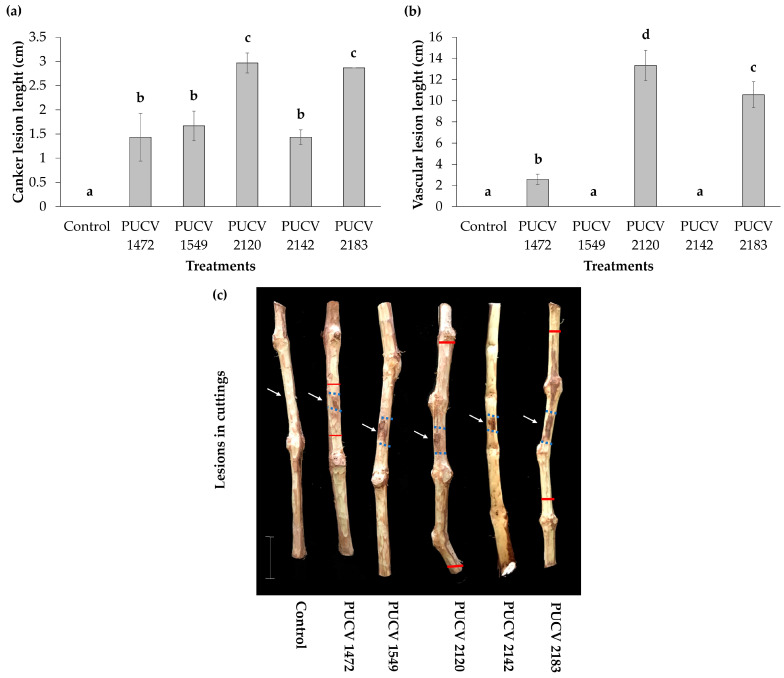
Severity of the damage of the symptoms caused by *D. seriata* on detached canes of two-year-old *Vitis vinifera* cv. Cabernet Sauvignon. (**a**) Mean canker lesion length. Different letters indicate significant differences after ANOVA according to Tukey’s test (*p* ≤ 0.05). (**b**) Mean vascular lesion length. Different letters indicate significant differences after ANOVA according to Tukey’s test (*p* ≤ 0.05). (**c**) Vascular and canker lesions on cuttings with activated isolates. The white arrow indicates the point of inoculation. The red line describes vascular discoloration. Lines over bars on graphs represent standard deviation.

**Figure 3 plants-12-02984-f003:**
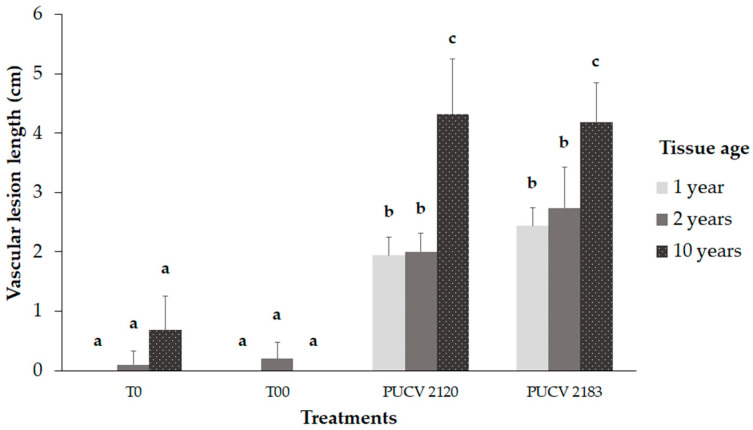
Damage severity of symptoms (vascular lesion length) caused by *Diplodia seriata* on tissue of different ages of *Vitis vinifera* cv. Cabernet Sauvignon. T00: without cutting or inoculation; T0: cutting plus sterile distilled water. PUCV 2120: *D. seriata* isolate. PUCV 2183: *D. seriata* isolate. Different letters indicate significant differences after ANOVA according to Tukey’s test (*p* < 0.05). Lines over bars on graphs represent standard deviation.

**Figure 4 plants-12-02984-f004:**
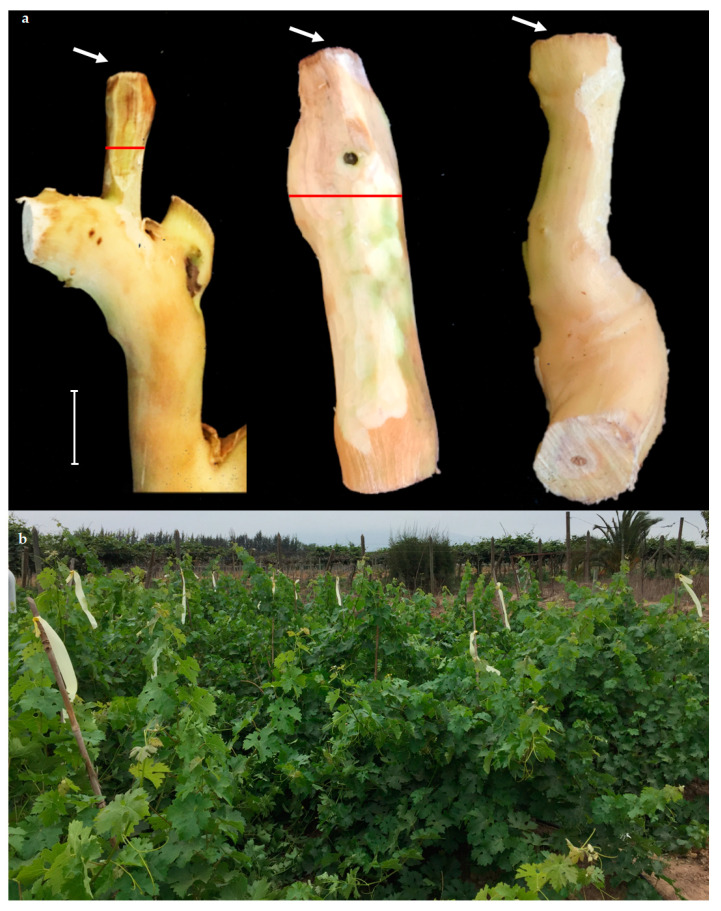
Symptoms developed on *Vitis vinifera* cv. Cabernet Sauvignon plants five months after inoculation with *D. seriata*. (**a**) Left to right: two-year-old tissue vascular lesion and 10-year-old tissue vascular lesion caused by the PUCV 2021 isolate, and 10-year-old tissue of noninoculated plants (T0). The white arrow indicates the inoculation point. Red lines delimit vascular discoloration. Scale bar = 2 cm. (**b**) General view of the trial in vines; the image was taken the day before evaluating the symptoms in the different treatments. Note the absence of foliar symptoms on the plants.

**Figure 5 plants-12-02984-f005:**
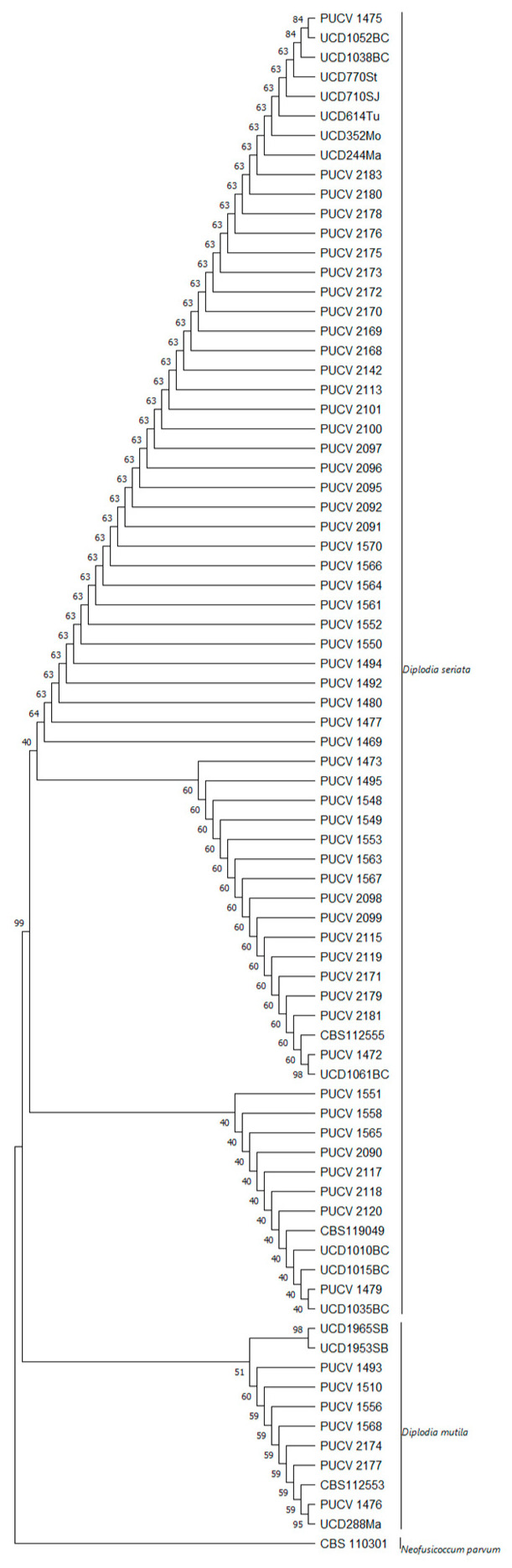
Phylogenetic tree of isolates of *D. seriata* based on maximum parsimony (MP) analysis of two loci (ITS and β-tubulin). Chilean isolates used in this work with PUCV code. MP bootstrap values are shown above the branches. *N. parvum* isolate CBS 110301 was the outgroup used. Values were obtained with Mega X software.

**Figure 6 plants-12-02984-f006:**
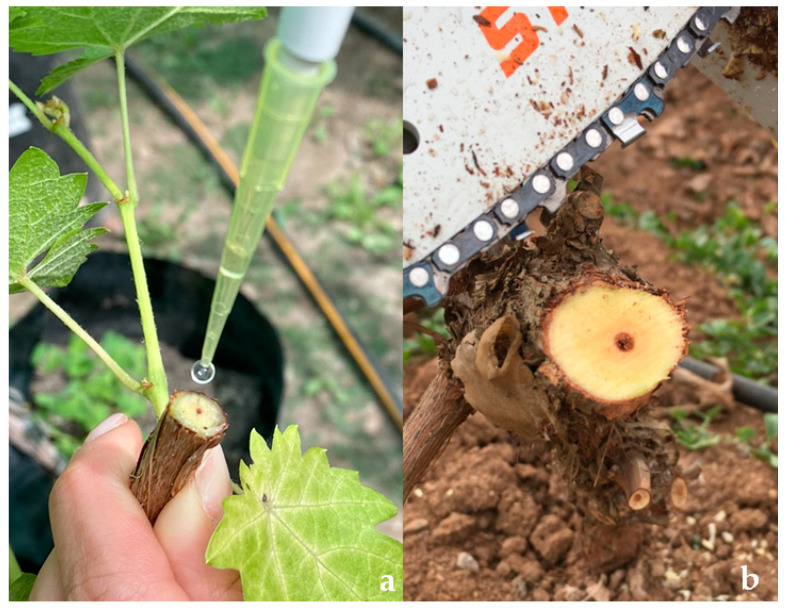
Inoculation with conidia solutions of *D. seriata* in grapevine plants cv. Cabernet Sauvignon. (**a**) Inoculation of 2-year-old shoots on fresh pruning cut. (**b**) Inoculation of 10-year-old wood (arm) freshly cut by chainsaw.

**Table 1 plants-12-02984-t001:** Chilean isolates of *Diplodia seriata* obtained from canker and vascular lesions of *Vitis vinifera* cv. Cabernet Sauvignon used for pathogenicity tests.

Isolate of *Diplodia seriata* *	Year of Collection	Locality Origin, Region of Chile	Access No. GenBank		
			ITS	BT	EF1-α
PUCV 1472	2010	Palmilla, O’Higgins	KM372581	KP762454	-
PUCV 1549	2010	Peralillo, O’Higgins	KM580514	KP762464	-
PUCV 2120	2018	Palmilla, O’Higgins	MT023573	MT063140	MT120827
PUCV 2142	2018	Batuco, Maule	MT023574	MT063141	MT120827
PUCV 2183	2018	Pencahue, Maule	MT023587	MT063154	-

* Larach et al. [5].

## Data Availability

Not applicable.

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
