# Peer review of "Higher Virulence of Diplodia seriata Isolates on Vines of cv. Cabernet Sauvignon Associated with 10-Year-Old Wood Compared to Young Tissue"

_plants, 2023, doi:10.3390/plants12162984_

Round 1

Reviewer 1 Report

Overall summary: Larach et al. compare the relative levels of virulence caused by Diplodia seriata on grape plants different growth spans. They find that certain strains can be more virulent on older trees compared to younger ones. While the manuscript provides a basic comparative analysis of age vs. pathogenicity, there are issues with the strains included in the experiments and the presentation of the results. See specific comments below.

Specific comments:

·         The introduction section does not cover the extent to which the pathogenic mechanisms of D. seriata have been studied at a physiological and molecular level. Some review of this should be covered here.

·         The structure of the methods section is not clear in terms of figures and tables being present there. The figures should be moved to the results section and should be described there, not in the methods section.

·         There is a conclusion section at the end of the methods. This should just be included in the discussion.

·         Line 75: The coloration of the tissue is a reference to figure 1C not 1B, this needs to be corrected.

·         Figure 1: Why were the detached shoot assays only conducted in 2 year old trees and not other ages? This needs to be included or explained.

·         Figure 1C does not have any labelling of the shoots and which strains they were infected with. It is hard to understand this image. Labels need to be added here.

·         PUCV2120 and 2183 have quite a high level of virulence observed in terms of vascular lesion even in the inactivated form. This was not highlighted in the results section. Similar trend with 2142 and 1549 in terms of canker lesion.

·         There are no statistical analysis mentioned in the description of figure 1A and B in the results section and only a physiological description. This needs to be majorly improved.

·         Figure 2 : There are no statistical measurements or comparisons of the lesion sizes in the berry grapes inoculated with various strains. Some measurements are mentioned in the results text, but this experiment should be described better in terms of the number of replicates included in the study and all the average measurements should be compared.

·         The structure of the results section should be made more sequential. Figure 1 is described a little at first then it jumps to figure 2 and back to figure 1. Each figure should be described in detail before moving to the next one. (Sections 2.1.1, 2.3.1)

·         Figure 3: Why were the other strains not included in this experiment? Only two strains are shown in the graph. If not this needs to be explained or the strains should be included.

·         Figure 4: In 4A, it is unclear what the scale of the red line is, it is vaguely described in the legend. This figure needs to be labelled with the inoculation conditions and plant stage within the image.

·         Which plants does figure 4B include? Is it all the plants in the study or specific ones, this needs to be labelled in the image or else the image can be omitted since it doesn’t provide any data. This should be replaced by images of plants inoculated with different D. seriata strains. Further, all strains should be tested in this experiment, not just 2120 and 2183.

·         Figure 5/Methods section related to this: Were the different strains used in the analysis part of a collection that was tested by PCR by this group or were these mined from NCBI or another database. The methods section does not describe this enough and the supplemental info section is missing. Not much info is provided about these strains.

·         The discussion section should include some information about the phylogenetic analysis in the study and how that relates to the evolution of the pathogen and the distribution of the disease.

Author Response

We appreciate your review.  We are pleased to send you  the revised version following the recommendations made. In the attachment we describe  the main changes introduced in the new version of the manuscript in order to respond to all the comments. We believe that the text was substantially improved following the suggestions.

Reviewer 2 Report

Comments and Suggestions for Authors

Manuscript ID: plants-2510580

The paper entitled “Higher virulence of Diplodia seriata on vines of cv. Cabernet Sauvignon associated with 10-year-old wood compared to young tissue” was carefully reviewed. This paper aims to test the pathogenicity of different isolates of D. seriata on detached canes and vine plants, compare the susceptibility of young and old wood of Cabernet Sauvignon to D. seriata infections, and contribute to the knowledge of the epidemiology of Botryosphaeria dieback in Mediterranean climates.

The present study focused only on one grapevine cultivar (Cabernet Sauvignon) and used one pathogen species (D. seriata), so the results may not be generalizable to other cultivars and pathogens. The authors only used detached canes and field inoculations, so the results may not reflect the natural conditions and interactions of the disease. Moreover, the study did not measure or report any physiological or biochemical parameters of the infected tissue, such as water potential, phenolic compounds, or defense enzymes, so the mechanisms of infection and resistance are not clear. These limitations should be taken into consideration in the present paper when interpreting the findings and applying them to other contexts.

Detailed comments:

Abstract

-          Add quantitative data to the abstract.

-          The abstract should also mention the limitations and future directions of the research.

Introduction

-          Line 36: Correct “dieback, and (BD) (BD)” by “dieback, and (Botryosphaeria dieback) (BD)”.

Materials and methods

-          The authors should describe how they performed the statistical analyses in a separate section, and provide more information on the distribution of the data and the tests used for each parameter. It is important to check the normality of the data before applying tests.

Results:

-          Add legends for figure 1c. Also, the presence of a scale bar in the image is required for measuring the lesion.

-          Label the letters and bars in figures 1a and 1b.

-          Add a paragraph to describe figure 1c in the text.

-          Delete the following sentences from the "results" section as they are already mentioned in the "materials and methods" section:

o   Lines 66-68 “Ninety days after inoculation with D. seriata isolates PUCV 1472, PUCV 1549, PUCV 2120, PUCV 2142, and PUCV 2183, the vascular lesions and canker lengths on detached canes 2 years old were measured”;

o   Lines 109-111 “Fifteen days after inoculation with D. seriata isolates PUCV 1472, PUCV 1549, PUCV 2120, PUCV 2142, and PUCV 2183 on berry grape cv. Red Globe, the diameter of lesions was measured”;

o   Lines 133-135 “Ninety days after inoculation with isolates of D. seriata PUCV1472, PUCV1549, PUCV 2120, PUCV 2142 and PUCV 2183 (previously inoculated in berries) in detached 2-year-old canes, vascular and canker lesion lengths were measured”;

o   Lines 152-154 “Five months after inoculation with D. seriata isolates PUCV 2120 and PUCV 2183 on tissues of three different ages (one-year-old, two-year-old and 10-year-old wood) of plants cv. Cabernet Sauvignon, the length of vascular lesions was measured”.

-          Remove the following subheadings from the results section as they are irrelevant and to avoid redundancy: “2.1.1 Damage assessment and recovery of pathogens”;

o   “2.2.1 Damage assessment and recovery of pathogens”;

o   “2.3.1 Damage assessment and recovery of pathogens”;

o   “2.4.1 Damage assessment and pathogen recovery”.

-          Figure 2b: “Internal damage of inoculated berry grapes, control berries (noninoculated) and berries isolated with recovery of D. seriata from inoculated berry grapes”. Do this caption in the order it appears in the photos.

-          Label the letters and bars in figure 3.

-          Lines 190-203 (2.5 Maximum parsimony analysis): This paragraph is too long and complex. It should provide some context and explanation for the phylogenetic analysis and its relevance to the study objectives and results, not just describe the methods and statistics. I recommend rewriting this subsection.

Discussion:

-          The discussion section is too long and contains too much background information. It should focus on interpreting and explaining the results of the study, not repeating the literature review. The authors should provide some comparison and contrast of the results with previous studies, highlighting the similarities and differences, and discussing the possible reasons and implications.

-          The first paragraph is too long and complex (211-252)!  

-          Lines 212: Replace “Diplodia seriata” by “D. seriata”.

-          Lines 213-216: Delete the following sentence as it is already mentioned in the introduction section: “Diplodia seriata is one of the most prevalent pathogens isolated from grapevines with symptoms of (BD), mainly in wine or table grape-producing areas with Mediterranean climates [5,6,15,17,23,25,26,28,36]”.

-          Lines 326 & 327: Replace “Vitis vinifera” by “V. vinifera”.

-          The following papers should be considered to enrich the discussion section:

o   ELENA, G., BRUEZ, E., REY, P., & LUQUE, J. (2018). Microbiota of grapevine woody tissues with or without esca-foliar symptoms in northeast Spain. Phytopathologia Mediterranea, 57(3), 425–438. https://www.jstor.org/stable/26675705;

o   Luque, J., Martos, S., Aroca, A., Raposo, R., & Garcia-Figueres, F. (2009). SYMPTOMS AND FUNGI ASSOCIATED WITH DECLINING MATURE GRAPEVINE PLANTS IN NORTHEAST SPAIN. Journal of Plant Pathology, 91(2), 381–390. http://www.jstor.org/stable/41998633

o   Bruez E, Vallance J, Gerbore J, Lecomte P, Da Costa JP, Guerin-Dubrana L, Rey P. Analyses of the temporal dynamics of fungal communities colonizing the healthy wood tissues of esca leaf-symptomatic and asymptomatic vines. PLoS One. 2014 May 1;9(5):e95928. doi: 10.1371/journal.pone.0095928.

o   Kraus, C.; Rauch, C.; Kalvelage, E.M.; Behrens, F.H.; d’Aguiar, D.; Dubois, C.; Fischer, M. Minimal versus Intensive: How the Pruning Intensity Affects Occurrence of Grapevine Leaf Stripe Disease, Wood Integrity, and the Mycobiome in Grapevine Trunks. J. Fungi 2022, 8, 247. https://doi.org/10.3390/jof8030247

o   Azevedo-Nogueira Filipe, Rego Cecília, Gonçalves Helena Maria Rodrigues, Fortes Ana Margarida, Gramaje David, Martins-Lopes Paula. The road to molecular identification and detection of fungal grapevine trunk diseases. Frontiers in Plant Science. 13, 2022. https://www.frontiersin.org/articles/10.3389/fpls.2022.960289

Conclusion

-          I recommend adding one or two sentences in the end of the conclusion section to mention the limitations of the study, and suggest some directions for future research.

The manuscript would really benefit from proofreading by an English editing service or a native English speaker as some sentences are very difficult to understand and there are many errors and typos.

Author Response

(The authors gave the same response as above.)

Round 2

Reviewer 1 Report

The authors have addressed the feedback. 

Author Response

Dear Reviewer
We appreciate the review, it contributed greatly to improve the manuscrip.

Best regards

Reviewer 2 Report

This revision has been considerably improved on the previous version. However, the authors did not take into account my previous comment: "The present study focused on a single grapevine cultivar (Cabernet Sauvignon) and used a single pathogen species (D. seriata), so the results may not be generalizable to other cultivars and pathogens. The authors used only detached canes and field inoculations, so results may not reflect natural conditions and disease interactions. In addition, the study did not measure or report physiological or biochemical parameters of the infected tissue, such as water potential, phenolic compounds or defense enzymes, so the mechanisms of infection and resistance are unclear. These limitations need to be taken into account in this paper when interpreting the results and applying them to other contexts". These issues should be addressed in the revised version to enrich the discussion section.

Other comments:

-          Lines 44-48: “The percentage of replicate trees in which the associated taxa clustered together in the bootstrap test (1000 replicates) are shown next to the branches. The MP tree was obtained using the tree-bisection-regrafting (TBR) algorithm”. Rewrite the complete sentence to avoid plagiarism.

-          Lines 58-61: Correct this sentence “In work carried out by Morales-Cruz et al. [36], it was determined that, in general, D. seriata has fewer genes associated with CAZymes enzymes, enzymes associated with secondary metabolism, active enzymes, and cytochrome oxidase 450 than N. parvum” as follows:

“The study by Morales-Cruz et al. [36], found that D. seriata has fewer genes associated with CAZymes enzymes, enzymes associated with secondary metabolism, active enzymes, and cytochrome P450 than N. parvum”.

-          Lines 174 & 271: The authors refer to tables S1, S2 and S3, which are not included in the paper.

-          Lines 196-199: “… to several factors, including the differential susceptibility of the vine varieties, the climatic conditions, age and tissue type of the host plant, the inoculation method used, the geographic origin of the isolates, and the different incubation periods of the experiments”. Rewrite the complete sentence to avoid plagiarism.

-          Lines 236-275: The paragraph is too long and complex. Please synthesize and simplify this paragraph.

-          Lines 263-265: “Martos et al. [65] demonstrated that D. seriata can produce secondary metabolites with phytotoxic properties, and it is possible that it has other mechanisms besides mycelial growth in the host that could be related to its virulence”. Rewrite the complete sentence to avoid plagiarism.

-          Lines 409, 414 & 418: Correct this “Tukey’s test (p = 0.05)” as follows: “Tukey’s test (p < 0.05)”.

-          Lines 416-418: The following sentence, describing the statistical analysis used to compare the field pathogenicity test, is still unclear: " The pathogenicity test in the field was distributed according to a completely randomized design, one plant was the experimental unit, and five replicates were used. Analysis of variance of two factors and Tukey's test were used for data analysis (p = 0.05)". The authors applied a two-factor analysis of variance. A one-way or two-way ANOVA? Which two factors did you take into account?

The manuscript would really benefit from proofreading by an English editing service or a native English speaker as some sentences are very difficult to understand and there are many errors and typos.

Author Response

Dear reviewer
We are very grateful for the review. It contributed to improving the manuscript considerably. Attached are details of the requested.

Best regards

Round 3

Reviewer 2 Report

This revision has been considerably improved compared to the previous version. The authors have satisfactorily addressed most of my concerns. I found a few small typos and some missing references in the text (Lines: 38, 40 & 307). I would recommend the manuscript for publication.